# Losing and Finding Braj: Commodification and Entrepreneurship in the Sacred Land of Krishna

Leena Taneja

Department of Humanities and Social Sciences, Zayed University, Dubai P.O. Box 144534, United Arab Emirates;
leena.taneja@zu.ac.ae

**Abstract:** Braj is a sacred place revered by Bengali Vaishnavas, followers of the bhakti sect of Vaishnavism, one of four branches of Hindu devotion. Followers of the sect worship the God Krishna, who it is believed manifested in Braj and carried out many divine feats and activities that are imprinted onto the land. Braj today is dotted with thousands of holy shrines, temples and natural places connected to Krishna. Devotees connect to Krishna through the landscape of Braj; it is where the transcendental and the physical realms meet. Braj has been transformed in a multitude of ways with the influx of money from Western sources, commercial enterprises and developers that wish to modernize and commercialize it for the new religious consumer. New infrastructure, condo developments and other changes illustrate both the challenge and the promise of modernity. This paper examines how these transformations are impacting the region of Braj.

**Keywords:** sacred landscapes; cultural heritage; Hinduism





## 1. Introduction

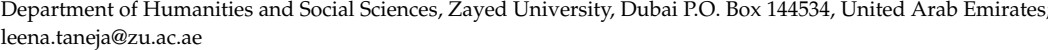

"When we go to the banks of the Yamuna and other lakes of Vrindavan or near Govardhan hill or the pasturing field, we see that the impressions of Kṛṣṇa's footprints are still on the surface of the earth. We remember Him playing in those places because He was constantly visiting them. When His appearance within our minds becomes manifest, we immediately become absorbed in thought of Him." (Prabhupada 1970, p. 59)

The sacred land of Braj, derived from the Sanskrit word *vraja*, is revered by Gaudiya Vaishnavas, followers who make up one of four branches of Hindu devotionalism or bhakti. Located thirty miles north of Fatephur Sikri and ninety miles south of India's capital, New Delhi, it is an internationally celebrated site where it is believed the young Krishna grew up, defeated demons and frolicked in the pastoral forests with his cowherd friends. For Vaishnavas, devotees of Krishna, these colorful mythologies provide a powerful mental map that inspires their creative imagination and reenacts the values and beliefs central to the bhakti faith. Referring to the sacred land of Braj, the scholar David Kinsley writes "Hindu spirituality is strongly geographical and involves learning how to read the landscape" (Kinsley 1998, p. 228). Kinsley's comment highlights how, in bhakti spiritual practice, geography and the mythology of Krishna are brought together so that reading the landscape through ritual practices such as touching, bowing, walking and singing is a powerful way in which devotees connect to Krishna's activities and form. Similarly, highlighting the construction of Braj's sanctity, Sugata Ray writes, "The natural environment of Braj was *given* liturgical significance equal to the elaborate rituals of icon worship" (Ray 2019, p. 13). What is significant about this quotation is the equation of the landscape with the icon itself—suggesting that worshipping the landscape is the same as worshipping the iconic image of the divine.

Braj is not unique and shares this eco-theology with other sacred Hindu sites in India. India's geographical landscape has been described as a divine hierophany. The sacred

landscape is imprinted with traces of divine stories and pastimes (*lilas or divine "play" or "sports"*) derived from Puranic literature dating as far back as the 4th CE. Pilgrims and devotees who visit the holy sites learn, recite and relive these stories through ritual performances and symbols tied to the land (Kinsley 1998; Haberman 1994; Eck 1981). For millions of pilgrims, the physical, imagined and spiritual engagement with the landscape is a powerful way to visualize, enact and reconnect to the divinity embedded in the land. This theology no doubt has ramifications on praxis. The Hindu religious experience is largely sensory, much of it focused on visualizing the deity. In Sanskrit, *darsan* means "to see and be seen by the divine image". But seeing and experiencing divinity is not limited to the anthropomorphic image of the deity worshipped in the temple or at home. For Hindus, access to the divine transcends images, icons or *murtis*. Material culture converges and coalesces with the natural environment, expanding the devotee's access to the divine through stones, trees, mountains, rivers and other natural phenomenon. Thus, a strong intersection exists between the human, spiritual and natural worlds. Myths and rituals help to manifest this intersection in manifold ways. Scholars have noted that divinity is embedded in the landscape in India through myths and *lila* (divine activities) that are imprinted into the land, surcharging it with divinity. Devotees touch the rocks, walk on the sacred dust, hike the mountains and bow down to the trees, stones and rivers, where the sacred is uniquely manifested. In this way, they revisit, relive and reenact those myths and divine *lilas* to gain access to higher realms of consciousness. In short, sacred landscapes are replete with religious symbolisms and rituals that are central to the facilitation of the devotee's religious experiences, solidifying values and identity within the community.

## 2. The Interdisciplinary Field of Ecology and Religion

The intersection of the environment and theology in Braj has been explored by scholars. One pivotal text worth noting on this subject is Lance E. Nelson's *Purifying the Earthly Body of God: Religion and Ecology in Hindu India* (2000). Speaking on Vaishnavism, Patricia Y. Mumme's article, "*Model and Images for Vaishnava Environmental Theology: The Potential Contribution of SriVaishnavism*", in this same book, was instrumental in showing me how the bhakti Vaishnava schools could contribute to a new intellectual discussion surrounding the ecological crisis. Another noteworthy contributor includes David Kinsley's article, "*Learning the Story of the Land: Reflections on the Liberating Power of Geography and Pilgrimage in the Hindu Tradition,*" which has been quoted above. The comparative essay seeks to look at how two different cultural pilgrimage sites, Australian aboriginals' view of their land and Braj, are linked together in their use of reading mythology into the landscape to uncover the story. As the landscape narrates the story, the pilgrim is taken on an internal and external journey that is deeply transformative. A more recent contribution to this interdisciplinary work by Sugata Ray's *Climate Change and the Art of Devotion* (2019) considers the "intersection of visual practices and large-scale transformations in the natural environment" (p. 8). He aims to "bring environmental humanities into conversation with South Asia's early modern and colonial art history" (p. 16). His work is particularly relevant to this paper because, unlike previous thinkers, his work provides a paradigm for decentering the idea of Braj from its essentialist form in which Braj is primarily viewed as a sacred place charged with divine immanent energy. In contrast to this dominant view, Sugata explores how environmental and social forces have shaped Braj's identity. His perspective advances the discussion by decoupling Braj from its mythological associations.

Sugata Ray's analysis can be further enhanced by viewing the landscape as a cultural landscape. A cultural landscape, by definition, is created, appropriated, organized and represented by human agency (Campo 1998). Such a view emerges from the idea that landscape is something modified through cultural processes. For a more pointed definition, "Cultural landscape is a place-specific ensemble of symbols, rituals, behavior, and everyday social practices that help in developing a shared set or sets of meanings" (Sinha 2006, 2014). This theoretical principle can be usefully applied to Braj. Sinha writes, for instance, that the ritual enactments performed in Braj "are place-making activities, marking sites and

leaving traces in the landscape palimpsest, making it a repository of collective memories" (Sinha 2014, p. 60).

Inspired by both Sugata and Sinha's perspectives on Braj, this paper seeks to de-privilege the mythological importance that has embodied the Braj landscape by paying closer attention to factors, both internal and external, that are commodifying Braj, such as through the astonishingly rapid rise in urban development spurred by an entrepreneurial spirit and the influence of capitalism. Looking at case examples like the new modern consumer/tourist in Braj and development projects, this paper contends that the transformation in Braj is layered and complex.

Logically speaking, if cultural landscapes are continually reshaped by the activities that take place within them, then changes in human activities, endeavors and values must impact the cultural landscape and the practitioner's experience that is tied to it. So, this paper asks: How are the interplay of these human processes shaping the architectures of Braj? How do larger global forces re-configure the lived experience of Braj as a representation of the divine form of Krishna? This paper contends that these processes—local and global—are not only altering the infrastructure of Braj but also the values and mindsets of practitioners. The symbols, rituals and interpretations surrounding the sacred landscape are slowly changing, as the dialectical relationship forged between the practitioner and the landscape is renegotiated in new ways to address the challenge of modernity. The hope is that by de-mythologizing Braj, at least in part, this analysis may offer a different, more nuanced view of its contemporary transformation. Furthermore, this analysis aims to illustrate that secularism, grounded in an alternative cultural context, allows us to understand the different ways in which modern culture materializes visually—or is produced, reproduced and transformed—and with them the making of religious sites and objects.

This paper will begin by foregrounding this discussion by taking a longer look at the early development of Braj in the common era and the efforts of by Caitanya Mahāprabhu (1486–1533), the founder of Gauḍīya Vaiṣṇavism, to revive the region in the sixteenth century. This discussion will help to anchor and contextualize more recent developments in the area. The second part of this paper will be devoted to examples drawn from fieldwork carried out by the researcher in 2009–2010 and from 2019 to the present. The section will also draw from Dr. Hawley's recent work in *Braj, Krishna's Playground: Vrindavan in the 21st century (2019)*, Dr. Sinha's article *The Sacred Landscape of Braj: India* (2014), and Eben Graves work, *"The Marketplace of Devotional Song: Cultural Economies of Exchange in Bengali Padavali-Kirtan."* Before proceeding, it should be noted that multiple Vaishnava communities thrive in Braj; however, this paper is limited to the Gaudiya Vaishnava sect due to the interest of the researcher.

## 3. Braj Lost: Braj Found

Throughout Braj's checkered history, the theme of paradise lost and paradise found is a discernable one. Many scholarly works have already explained the history of Braj and its development. This section will highlight some of the varied ways that Braj has been constituted over time by highlighting the multiple holy figures, literatures, visionaries, entrepreneurs and political actors that have played in a role in constituting Braj over time. I argue that the theme of lost and found is a useful way to discuss these transformations—perhaps even to suggest that it is a metaphor for the devotional journey that the devotee endures in finding Krishna. After all, if the land is equated with Krishna himself, then finding and losing Him is part and parcel of their spiritual quest. In no way is this treatment exhaustive since it is intended to suggest the possible human responses that flowed into and left its mark on this region.

In his book, *God: A Human History of Religion*, Reza Aslan traces the idea of a "humanized God"—one that looks like us. Beginning with the early neolithic period, his argument is that humans have fashioned God in their form and based on their needs to survive as a species. The thesis is provocative and though he applies his theory to the Abrahamic religions, his thesis can be usefully applied in different cultural and historical contexts. In

the Neolithic period, the "process of transforming the earth to our advantage brought with it a whole new set of values and behavior norms as well as a new collection of myths to help us make sense of the changed world we inhabit" (pp. 59–60). He notes that the movement from hunting to farming in other regions of the world coincided with a new religious sensibility that divinized the earth. After all it was the earth—and the harvest—and not the hunt that humans were dependent on for their survival. Speaking of this transition in the Neolithic period, Reza Aslan notes, "We stopped praying for help with the hunt and prayed instead for help with the harvest" (Aslan 2018, p. 59). Divinity transfers and is reconfigured as humans evolve from one mode of survival to another. He continues that it was around this time that the concept of an "immolated deity" first arose—the god who does and is dismembered and from whose body creation springs (Aslan 2018, p. 60). This harkens back to the idea of purusha, the primal man, in the Vedas, but it also resonates with the topographical view of Krishna articulated by Narayana Bhatt, the seventeenth-century biographer, who wrote in *Vrajabhaktivilasa*, "The Braj mandala is an essential form of the Lord consisting of organs and limbs. Mathura is his heart; Madhuban is his navel; Kamudban and Talban are his breasts; Vrindavan is his brow; Bahulaban and hamaban are his two arms; Bhandiraban and Kokilaban are his two legs; Khadiraban and Bhadrikaban are his two shoulders; Chatraban and Lohaban are his two eyes; Bhadaban are his two ears; Kamaban is his chin; Triveni and Sakhikubaban are his two lips" (Haberman 1994, p. 126). Aslan contends that "such mythologies better coincide with the birth, death, and rebirth of our crops, they fostered a more intimate relationship with the divine" (Aslan 2018, p. 60). Looking closely at Krishna's mythologies, themes of climate, land and fertility are evident. For instance, the story of Mt. Govardhan centers on a thunderstorm. The story of Trinivarta similarly is about a whirlwind that comes in the form of a demon. The story of Kaliya, the serpent demon, highlights the pollution of Yamuna. All themes, one might extract, that imbibe an agricultural sensibility.

What Reza does is help us to look more carefully at the connection between agricultural societies and the embodied conception of God linked to the land. If this happened in other religions, it may well have been a factor that created the place-based mythology of Krishna that embeds itself in Braj over a sustained period.

Historical accounts of Braj dating back to 400 CE describe the region as "a thriving Buddhist pilgrimage center with over twenty monasteries and three thousand monks" (Ray 2019, p. 8). Archeological data of sculptures also reveal that in the early centuries of the Common Era, there were Hindu and Jain temples being built during this period (Ray 2019, p. 9). It was by all accounts an ecumenical, thriving religious center. Adding to this eclectic religious scene, archeological finds in the Mathura area attest to the "widespread worship of nature deities, evidenced by the numerous images of local nature spirits known as yakshas, nagas (a form of snake god), and tree spirits" (Packert 2010, p. 3). Cynthia Packet also notes that ancient regional ancestor and hero cults with their themes of embodiment in human form and connections to royalty and heroism are another component that influenced the Krishna cult (p. 3).

We now proceed to the 8–9th centuries, when transformations begin to happen to the form of Krishna. The *aishwarya* (monarchial) form of Krishna associated with the ritualized, orthodox form of vaidhi bhakti is heavily influenced by the Alvars, South Indian mystical poets. They inject the kingly Krishna with some emotionalism and turn him into the *madhurya* form (amorous) divinity (Hardy 2014). This new emotional variety of bhakti makes its way to West Bengal in Eastern India, where it is later theologized and spread by Caitanya Mahaprabhu and his associates. According to this new love theology, Krishna can be found not through strict rules and regulations but unconditional, passionate love. Caitanya brings this emotionally charged form of bhakti to Braj around 1514. By the sixteenth and seventeenth centuries, the Jain and Buddhist structures are long gone. In their place, four red sandstone temples are erected in Braj that begin the establishment of Krishna worship in Braj (Ray 2019, p. 10). The area sees a fervor of religious pioneering activity in the late fifteenth and early sixteenth centuries, when many

sites are rediscovered and reclaimed by Vaishnava pioneers under the guidance of *Caitanya Mahaprabhu* and *Vallabhacharya* (1479–1531), the founder of the *Nimbarka* Vaishnava sect. These new structures bring with them a new paradigm of bhakti, new symbolisms and vocabularies that take root in Braj. Central to this new variety of bhakti is the idea that Krishna is *one* with Braj. But this is all lost when the area is annexed by the early Mughal ruler, Akbar.

Twists and turns in this narrative only continue today. A religious community in Northern California called New Braj a Western transplantation to the United States. The old Braj is "lost" and a new one is erected in its place.

## 4. Braj: The Embodied Landscape of Krishna

According to the Gaudiya tradition, Braj is no ordinary place. According to the Bhagavad Purana, it is believed that Krishna descended on Earth and performed lilas (divine activities) in Braj with his divine associates. Braj is considered a replica of the divine abode of Krishna on Earth. Spiritually, Braj is not considered different from Krishna's own divine form, while temporally these divine pastimes continue to be played out on a spiritual dimension.

Harkening back to a type of Platonic dualism, Braj is often portrayed in both imagery and theology as two contrasting realms: the supramundane and the profane. The supramundane is the spiritual world inhabited by Krishna and its divine associates who continually perform lilas or in this numinous world. This realm is permanent, full of bliss and existence. The material or profane realm, on the other hand, is one of physicality and temporality. The spiritual world of Krishna informs and shapes the physical world of Braj in concrete ways. For devotees, the purpose of many spiritual practices in Braj is to cross over from one realm to the other, to move beyond and through the material realm into the spiritual world of Krishna and his divine associates.

From a practitioner's viewpoint, visiting Braj is a spiritual practice (*sadhana*) that is both an imaginative and a sensory experience wherein the devotee rediscovers the sites by hearing, retelling, reimagining and re-enacting Krishna's activities that took place in specific holy sites in Braj. These sites are sacred nodes surcharged with spiritual presence that ignites devotional love (*bhakti*) in the heart of the devotee. The landscape, Kinsley explains, has an implicit structure. He writes, "Hindus are preoccupied with learning the story of the land. Hindu pilgrimage involves a self-conscious attempt to cultivate a rapport with Indian geography that establishes, reaffirms or transforms one's religious identity" (Kinsley 1998, p. 228).

While the whole of India is considered the symbolic form of Bharat Ma or the goddess, Braj itself has been connected to specific parts of Krishna's body. Learning the land of Braj involves traversing the divine body made in geographical form. In short, Krishna is Braj, and Braj is Krishna (Haberman 1994, p. 127). The devotee's goal is to unveil the implicit, spiritual structure embedded in the landscape through ritual performances. I will explain a few of these ritualized activities below to illustrate how the devotee perceives and re-imagines the story in Braj. These ritual actions have been described by Dr. Sinha as a form of place making and provide an excellent way to understand how these practices contribute to our understanding of Braj as a cultural landscape.

*Chirghat* is a special place in Vrindavan where Krishna stole the clothes of Radha (his divine consort) and the Gopis, her friends, while they were bathing in the Yamuna River and hid them in the branches of a kadamb tree (indigenous tree). Pilgrims today visit *Chirghat* to hear the story and tie a piece of cloth to one of the branches of the tree. "This is a gesture of taking on the role of a Gopi in the effort to appease Krishna. Thus, the performance of rituals plays a significant part in reproducing a sacred landscape that exists in mythology" (Shinde 2010, p. 140). Another example is provided by Dr. Sinha, who examines the rituals associated with Govardhan Hill and the Yamuna River. Both sites are the most celebrated examples of natural archetypes of Hindu mythology and are the most ancient and sacred sites. They appear prominently in Krishna myths and are

celebrated through visual imagery in texts, paintings, sculptures and poetry. The story of Govardhan is drawn from the *Bhagavat Purana* (Book 10). In this story, the denizens of Braj are preparing for a great feast for Indra, king of the gods, but Krishna encourages the inhabitants to give their offerings instead to Govardhan Hill, which is responsible for giving them much needed food and shelter. Since Braj was a pastoral society, the shift from worshipping King Indra to the natural form of Govardhan is noteworthy. Indra, of course, is furious, and rains down on the inhabitants of Braj for eight long days. The storm is so severe that Krishna finally picks up Govardhan hill with his little pinky and raises it above the inhabitants like an umbrella to give them shelter. The moment is iconic and captured in many dioramas and dramas depicted in Braj. Today, the mountain is a low-lying hill stretched over 23 km. The mountain is now built up with temples, shrines, ghats (bathing platforms) and ponds. It is traversed by pilgrims who ceremoniously walk or prostrate themselves as they circumambulate its sacred boundary. In circumambulating the mountain, devotees reclaim the sacred mountain of Govardhan and recollect the myth of Govardhan, which illustrates how Krishna gave shelter and protection to the residents of Vrindavan. Such actions demonstrate that place making in Braj is centered on imagining, enacting and reclaiming features of the cultural landscape.

Yamuna River is a sacred river that appears in multiple Krishna myths found in the *Bhagavad Purana.* The river is a ubiquitous witness to most of Krishna's divine activities in Braj. For instance, it is said that the River Yamuna appears to participate and facilitate the manifestation of Krishna's divine *lilas* on Earth. She is considered a necessary actor in the unfolding of his manifold stories in Braj. In one poignant portrayal, in Krishna's birth story, Vasudev, the father of Krishna, carries the baby Krishna across the River Yamuna to protect him from being killed by his uncle, King Kamsa. The river calms, allowing Vasudev to accomplish his task, while a serpent shelters the father and child from the torrential rain under its hood. The scene is depicted in countless popular artworks and reenacted in dramas and plays during the festival of Janamastami, which celebrates the birth of Krishna. In his comprehensive book, *River of Love in an Age of Pollution: The Yamuna River of Northern India*, Haberman (2006) presents the river as a goddess flowing with liquid love. In his vivid descriptions, Haberman intermingles the geographical, mythological and physical features of the river, highlighting how myth sculpts the physical and sacred depictions of the river today and how devotees' ritual performances reinstate her significance. Communities and individuals worship the Yamuna and rely on her for sustenance. She is worshipped for her life-giving properties as the mother goddess. For instance, Yamuna *aachaman* (rinsing the mouth with holy water) at *Vishram ghat* (bathing platform) is considered the surest way of salvation, and pilgrims carry its holy water in sealed vessels for religious functions. Yamuna *poojan* (worship) is considered a must for candidates contesting elections in Mathura. On all auspicious festivals, pilgrims first take a ritualistic dip at the ghats before paying obeisance to the deities in the temples. Numerous examples such as these illustrate the confluence of myth, ritual and place making in Braj.

## 5. Cultural Commodification: Restoring Braj

Consistent with its historical past, the Braj of today is being restored to address a host of challenges that are impacting these indigenous practices. State government initiatives, green activists, urban developers, social activists, non-government organizations (Braj Foundation) and multi-national religious organizations (the largest known as Iskcon) are collectively shaping the new Braj. Braj is changing rapidly, with new projects and proposals being introduced to alleviate congestion to major temples, provide safer access for pilgrims and address pollution and deforestation and the loss of lakes and heritage sites due to damage caused by misuse, poor upkeep and misappropriation of the land by commercial enterprises. In this section, I will focus on current and future development projects in the region and untangle the web of competing interests and parties that are involved. I argue that this is not a black/white picture. Responsibility cannot be hoisted on any group or organization. There are many different motivations and interests involved in these projects.

Balancing the conservation and restoration of these heritage sites with the demands of modernity, such as accessibility, safety and the needs of the new religious consumers, has not been easy. Material in this section is drawn from the researcher's ongoing fieldwork. This ethnographic fieldwork has involved interviews with conservationists and developers, studying new projects and initiatives and first-hand observations of renovation projects. The research presented here is based on initial findings. I also draw on Dr. Jack Hawley's book, *Krishna's Playground: Vrindavan in the 21st Century* (2020). Analysis has been helped by the research on cultural landscapes provided by Amita Sinha, and Eben Graves' work on devotional songs and cultural economies has been helpful to explore the commercial aspects of this discussion.

During my fieldwork, I came across several non-government agencies working on various conservation and beautification projects in Braj. The Braj Foundation (www. brajfoundation.org, accessed on 1 December 2022), whose chairman is Vineet Narain, an eminent journalist, is a non-profit organization established in 2005. According to their website, the organization's purpose is to "restore and revitalize the environmental and cultural heritage of Braj". It has completed several projects on key sites in Mathura, Vrindavan, Govardhan and Barsana. One such project that I visited in Vrindavan is Seva Kunj. Discovered in 1590, the place is famous because, according to the Bhagavad Purana, the site is where Krishna performs his *rasa lila* or night dance with Radha and the gopis. The enclosure is a large, low-lying grove made up of 16,108 *tulsi* trees, a sacred plant, which I was told "change into 16,108 gopis at night so Krishna can carry out his mystical dance" (R. Murali, personal communication, 20 May 2011). I witnessed the site's condition prior to its refurbishment in 2011. The site was badly dilapidated, overrun by monkeys, and the concrete enclosure surrounding the grove had aged over time with little upkeep. In the interior of the grove is a temple known as Rang Mahal, with vivid depictions of Krishna and Radha's nightly escapades. Every evening, devotees worship at the temple to set the stage for Krishna's divine dance, setting out flowers and other accessories that the divine couple may need. The landscape is animated by myth and devotees relive the story every evening with lively chanting and puja. *Pujaris* (priests) even dress the deities for the rasa lila dance. Mythical connections are everywhere in Seva Kunj. It is said that the trees are low-lying because they are bowing down to the divine couple. But over the years, overgrown bushes, trees and pesky monkeys that harass devotees in the evening as they wind their way to the temple have emerged as physical and environmental challenges. Speaking with volunteers of the Braj Foundation about their refurbishment efforts, I learned that the *parikrama marg* or walk to the interior temple had been covered with a "large grill overhead to protect devotees from monkeys and the creepers". The unruly grove made up of creepers has a "new irrigation system and improved soil" (M. Gupta, personal communication, 13 August 2011). The wall that enclosed *Seva Kunj* and surrounds its perimeter was restored and the *Lalita Kunj*, a small lake within the enclosure, has been cleaned and beautified.

The beautification of *Seva Kunj* is one illustration of how non-profit organizations such as the Braj Foundation, funded and supported by Indian industrialists, are acting to restore and maintain the sacred landscape, with careful attention paid to its history and heritage. The work was described as "*seva* or service: a form of *bhakti* in action" (V. Narain, personal communication, 17 September 2011). I was also told that the service is a continuation of Caitanya Mahaprabhu's work, who recovered the lost sites in Braj. The Foundation similarly states that their *seva* efforts are reclaiming the sacred sites of Krishna's stories and saving them so that they can continue to surcharge the landscape. Sinha writes, "Conservation of heritage entails protection of the cultural landscape of narrative place markers, relics, and other commemorative structures that are mnemonic devices for keeping the place-bound traditions alive" (Sinha 2014, p. 70). In short, by connecting their efforts to the historical narrative of Chaitanya, the Foundation gains credibility and affirms the value of Braj's mythical past.

Some scholars have been quick to reject these kinds of initiatives, however, claiming that they cause a dissonance between the imagined landscape and the real experience. For

example, Dr. Sinha notes that the rise in religious tourism (five to seven million annually by some accounts) cannot be sustained by the landscape and has led to alternatives and modifications in religious behavior. For example, rather than walking, which heightens the tactile experience, pilgrims now choose to use cars, buses and other forms of transportation to quickly move from one sacred site to another. But, she notes, these conveniences prevent them from experiencing the landscape directly (Sinha 2014, p. 70). Sinha lays some of the blame for this disconnect on "insensitive development" (Sinha 2014, p. 59). Her argument is that when the state government of Braj meddles in how the land is developed, preserved and conserved, these measures are often ill-formed about the natural and spatial archetypes grounded in indigenous models. Alternatively, she suggests that "Site planning and management should take into account what is today considered non-essential knowledge—the language of myths, hidden meanings of rituals, and sanctity attributed to nature evident in everyday practices—so that a different paradigm for solving complex problems that defy standardized solutions can emerge" (Sinha 2014, p. 73). While there is some merit to Sinha's arguments, her views regarding Braj at times appear overly romanticized and idealistic.

She is not alone. There is stiff resistance from local critics as well, who maintain that any changes to the original character of Braj undermine its efficacy as a sacred site. Social activists, green activists and organizations like the Friends of Vrindavan, an ancillary of Iskcon that is devoted to "welfare activities concerned with the public good in Braj" (www.friendsofvrindavan.org, accessed on 1 December 2022), complain that the purity of the site is compromised when social utility is prioritized over the inherent sanctity of the site. A case in point is two new development projects approved by the Uttar Pradesh state government and its chief minister, Yogi Adityanath. One project is the development known as a heritage city near Mathura, which promises to be a city that "showcases the ethos of the Braj region" (Dev 2023). The new township will have "high-end tourist facilities, hotels, waterbodies, recreational areas" (Dev 2023) and it will develop the whole of Braj as a tourist-friendly region. The project is set to be completed by 2031. Several other approved projects aimed at improving accessibility include new expressways including the Chaurasi Kos yatra route and the Vrindavan–Mathura rope road. These latter initiatives, critics say, will alter the winding streets of Vrindavan. Jagananath Podder, a member of Friends of Vrindavan, stated, "The narrow Kunj galis (small streets) of Vrindavan are famous. Any widening after demolition of antique structures, will be a blow to the heritage character of Vrindavan, the soul of Sri Krishna-Radha Bhakti movement" (Podder, personal communication, November 2022).

In his new book, *Krishna's Playground: Vrindavan in the 21st Century*, Dr. Hawley outlines the changes occurring in the past ten or fifteen years in the region. He remarks that there are two camps in Vrindavan: the Futurists, those who think Vrindavan must develop, and the Protectors, who are worried about the impact of this development on Vrindavan. The state government is part of the Futurists, who are advancing new development projects aimed at improving the infrastructure but unfortunately have a poor understanding of the region and its traditions. An oversimplification, I think, but a binary that frames his main argument.

Developments in this region include new temples, dioramas and condo developments that are changing the landscape in immeasurable ways. A superb example comes from the Iskcon Bangalore Temple, which split from the parent organization of Iskcon several years ago. This new organization has designed a new 70-foot skyscraper temple called the Sri Vrindavan Chandrodhaya Mandir in Vrindavan that will be constructed just on the outskirts of the town. This tower has plans to re-create Yamuna River, calling it "Yamuna Creek", a more purified version that will allow people to bathe in its waters. The terminology and depictions of this new Yamuna River highlight the modernization that is prevalent in this region. Architects of this new temple have also envisioned a theme park in the temple, which will include rides inspired by Krishna's mythology, like the Trinavarta ride, which is based on the story of the whirlwind demon, Trinavarta, who was sent by the evil Kamsa to kill Krishna. In the *Bhagavad Purana* story, Krishna is pulled up into the clouds by the

mighty demon, but Krishna drags the demon down with his weight and kills him. The gamification of Krishna's mythology illustrates how religious and commercial enterprises are merging to change the idyllic landscape that once represented Vrindavan. The New Vrindavan that Dr. Hawley describes is like a mall complete with an anchor store, which, in this case, is the Iskcon Mandir. The surrounding area has been transformed in the past few decades with new temple and condo developments that have a kind of theme-park feel to them. The Vrindavan Chandrodhaya Mandir is designed to tower over older temples, competing for size and popularity. Similarly, along the main Chatigar highway, one now can witness large sculptures depicting Hindu gods, such as a 52-foot image of Hanuman, the monkey god, and Vaishno Devi. These huge structures can be seen from miles away and attract hundreds of pilgrims and tourists. In this way, the Vaishnavism of the town is slowly ebbing away as the town develops and is being eclipsed by a kind of Pan-Hindu form of Hinduism.

To support this spiritual infrastructure, there are plenty of condo developments as well that have transformed the area and that cater to the growing demand for long-term housing. These condos are primarily sold to wealthy middle-class Delhiites, foreign buyers and investors or those wishing to retire in Vrindavan. To attract clientele, western amenities are being included, such as pools, security, grocery stores, a gym and a children's playground. These developments are emerging all over Vrindavan, transforming the area so that it looks like a suburb of Delhi and not a holy site. Such conveniences and amenities are antithetical to the bhakti pilgrimage practice, which encourages austerity and self-sacrifice to bring one closer to Krishna and help one detach from worldly life. Contrary to these beliefs, the condo developers see an opportunity to deliver Krishna in a new package that is modified and monetized for the new devotee consumer.

## 6. Analysis: Material Culture and Commodification

Analyses from Eden Graves' work on devotional songs and David Miller's book on material culture raise issues of commodification and exchange value, which can be useful when applied to this discussion on Braj. Graves examines how devotional musical styles are transforming due to advertising, media production and promotional techniques. Critics, however, such as kirtan instructors, journalists and scholars, suggest that these professional kirtaniyas are guilty of changing the musical style to attract new audiences. They view these changes as a form of crass commercialism (Graves 2017, p. 52). One can see the parallels in this argument against the developers of Braj, who, critics argue, are re-packaging Braj to entice new consumers, watering down the authentic experience of Braj and erasing the link between the sacred land of Krishna, and the devotees and pilgrims who visit this holy place (Hawley 2020). According to Graves, two social values are associated with padavali kirtan—egalitarian promise and mass education (Graves 2017, p. 57). But the "processes of economic exchange are thought to undermine these social values" (Graves 2017, p. 58) and are being eroded in the face of economic exchange (Graves 2017, p. 59). Graves' discussion on kirtan draws attention to the economics of cultural objects and events in a religious context. He writes that "promoting an egalitarian social structure and mass education were the clearest markers of a kirtan's use-value in solidifying Bengali social identity. But in the discourse of commodification the social value of padavali-kirtan—its use-value—is compromised as musicians assign a price to performances—its exchange value" (Graves 2017, p. 59). Likewise, we can argue that Braj's use-value is Krishna's immanent divinity, accessible through the embodied experience of the land. This is compromised when it is repackaged and commodified to attract new consumers—its exchange value. The rise of exchange value results in the loss of use-value (Graves 2017, p. 59).

Advancing this argument further, we can apply David Miller's concept of objectification.

According to Miller, the concept of objectification refers to the way in which subjects and objects are mutually constituted. It is only when subjects externalize aspects of themselves in objects that they become aware of their own being. Next, since the externalized object itself possesses aspects of the subject, the latter reidentifies with the object, seeing

themselves in the object like a mirror. It is in this play of differentiation and identification that the subject and the object are reciprocally constituted (Miller 1994). With this, Miller invites us to consider the objectification of Braj that performs a certain role in culture and human relations. Braj is not an autonomous object but reciprocally participates in an interplay of difference and identity with multiple subjects. Braj constitutes and is constituted by subjects who construct, reproduce and circulate them. As a form of material culture, Braj holds certain properties and tendencies and is produced and reappropriated into new forms by the consumer. Subjects use these objects of consumption to negotiate new social frameworks imposed on them. Miller stresses that it is important to comprehend how the individual is given new meanings and is using and transforming these goods. In the case of Braj, it is evident that many individuals are imposing meanings and transforming it. This reconstitution has continued throughout its long history of loss and recovery.

## 7. Conclusions

This paper sought to de-privilege the mythological importance that has dominated discourse on Braj by re-situating Braj within the wider historical, social and economic processes that have shaped its identity. At the center, Braj has come to be equated with the divinity of Krishna and his earthly pastimes as set forth in the Bhagavad Purana. Finding Krishna for Vaishnavas entails active engagement with the cultural landscape. The cultural memories and ritual enactments sanctify the land and are transmitted from one generation to another. "Practices create ritual enactments that shape a dynamic, continuously evolving cultural landscape" (Sinha 2014, p. 66). The landscape is seen, felt, tasted and inscribed on the body through walking, bathing, singing, dancing and storytelling to create personal and collective place memories (Sinha 2014, p. 66).

Early historical accounts attest to a plethora of religious influences, visionaries and sects that retrieved and reconstructed Braj. Today, there is an astonishingly rapid rise in urban developments spurred by an entrepreneurial spirit and the influence of capitalism on the region. New expressways, refurbishment projects and urban developers are impacting the landscape, some say for the worse. There are multiple competing opinions, interests and motivations that are shaping the region, including the state government, local NGOs, sectarian groups, environmental and social activists and the new, modern religious consumer of Braj. Some say that the biggest loss is the experiential connection to the land and the mythical narratives that give it meaning. Applying analyses of commodification and material culture has served to bring a more critical eye to the discussion, wherein Braj is located in a web of frameworks that constitutes its meaning and significance.

**Funding:** This research received no external funding.

**Institutional Review Board Statement:** Not applicable.

**Informed Consent Statement:** Informed consent was obtained from all subjects involved in the study.

**Data Availability Statement:** No new data was created.

**Conflicts of Interest:** The author declares no conflict of interest.

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
