# Peer review of "Losing and Finding Braj: Commodification and Entrepreneurship in the Sacred Land of Krishna"

_religions, doi:10.3390/rel14050643_

Round 1
Author Response
- Bibliography revised
- Braj has been contextualized with historical background provided using new material.
- Pipeline examples removed and new empirical data from researcher used.
- South Asia secularism removed
- The work of such scholars as Cynthia Packert, Sugata Ray (art and ornament), Eben Graves (musical practice) have been added
- Justification provided for focus on gaudiya Vaishnavism
- Deleted Gaudiya Vaishnava intro section
Reviewer 2 Report
I really appreciated that a paper was written on this very interesting topic. I would recommend the article to be revised for its point(s) to be clearly demonstrated.

Author Response
1.Greater attention to critical analysis of the topic to avoid naive statements.
2. Climate Change and the Art of Devotion, Geoaesthetic in the Land of Krishna 1550-1850, by Sugata Ray (2019) used in the paper.
3.Early history of Braj summarized for historical background
4. Stressed the complicated nature of the transformation of Braj by not blaming any particular group. I specifically name key political persons and organizations.
5. Remove case examples like the pipeline example etc and added the researcher's own fieldwork with greater details.

Round 2
